# Characterization of Deep Learning-Based Speech-Enhancement Techniques in Online Audio Processing Applications

**DOI:** 10.3390/s23094394

**Published:** 2023-04-29

**Authors:** Caleb Rascon

**Affiliations:** Computer Science Department, Instituto de Investigaciones en Matematicas Aplicadas y en Sistemas, Universidad Nacional Autonoma de Mexico, Mexico City 3000, Mexico; caleb@unam.mx

**Keywords:** speech enhancement, online applicability, real-time factor

## Abstract

Deep learning-based speech-enhancement techniques have recently been an area of growing interest, since their impressive performance can potentially benefit a wide variety of digital voice communication systems. However, such performance has been evaluated mostly in offline audio-processing scenarios (i.e., feeding the model, in one go, a complete audio recording, which may extend several seconds). It is of significant interest to evaluate and characterize the current state-of-the-art in applications that process audio online (i.e., feeding the model a sequence of segments of audio data, concatenating the results at the output end). Although evaluations and comparisons between speech-enhancement techniques have been carried out before, as far as the author knows, the work presented here is the first that evaluates the performance of such techniques in relation to their online applicability. This means that this work measures how the output signal-to-interference ratio (as a separation metric), the response time, and memory usage (as online metrics) are impacted by the input length (the size of audio segments), in addition to the amount of noise, amount and number of interferences, and amount of reverberation. Three popular models were evaluated, given their availability on public repositories and online viability, MetricGAN+, Spectral Feature Mapping with Mimic Loss, and Demucs-Denoiser. The characterization was carried out using a systematic evaluation protocol based on the Speechbrain framework. Several intuitions are presented and discussed, and some recommendations for future work are proposed.

## 1. Introduction

Speech is captured in a real-life scenario along with noise and ‘interferences’ (other sources that are not of interest). The captured audio signal can be considered as an additive mixture of several audio ‘streams’, one of which is the original stream of the target speech. To this effect, speech enhancement is the removal of such noise and interferences from a given audio signal, such that only the target speech remains in the resulting enhanced audio signal. It has recently grown in popularity thanks, in large part, to the recent surge in performance of deep learning-based models [1]. Speech enhancement is closely related to ‘speech separation’ [2], since they bare similar objectives. However, in this work a distinction is made between the two, since the latter aims to separate all of the speech streams in the audio signal into their own audio separate ‘channel’, while the former aims to “separate” only the target speech. Still, both sets of techniques share several frames of reference, specifically, how to model speech and separate it from other audio streams in the mixture.

The types of speech-enhancement techniques can be divided in several ways, but the one relevant to this work is in the way the captured input audio signal is being fed to the technique. ‘Offline’ speech-enhancement techniques are fed the entire audio recording and, correspondingly, estimate all the target speech in one go. This type of techniques are of interest in applications, such as security by voice [3], audio refinement for musical recordings [4], and movie soundtrack enhancement [5]. ‘Online’ speech-enhancement techniques are sequentially fed audio segments of the captured input audio signal through time (usually coming directly from a live microphone) and, correspondingly, aim to estimate the target speech in each audio segment, which can then be outputted directly to a speaker, with some expected latency (which is the time difference between the moment the audio segment is fed to the technique and the estimated target speech in that audio segment is reproduced to a speaker). These type of techniques are of interest in applications, such as real-time automatic speech recognition [6], sound source localization in service robotics [7], hearing prosthesis [8], mobile telecommunication [9], and video-conferencing [10].

Although there have prior important and thorough comparisons between speech-enhancement techniques [11,12,13,14], they were carried out evaluating the techniques in an offline manner. As far as the author of this work knows, there has not been a comparison between speech-enhancement techniques with a focus on online applicability.

To this effect, the objective of this work is to characterize a representative sample of online speech-enhancement techniques, chosen because of their ease of replicability, online viability, and popularity. The characterization described in this work is based on a systematic evaluation, using a synthesized evaluation corpus by ‘disturbing’ a sub-set of recordings from the LibriSpeech corpus [15]. These disturbances were (1) adding noise at varying amounts of signal-to-noise ratio; (2) adding varying amounts of interferences with varying amounts of signal-to-interference ratio; and (3) adding reverberation at varying amounts of scaling factors. The ‘online’ aspect of this characterization was carried by feeding the techniques a sequence of audio segments extracted from the synthesized recordings, and then concatenating the results. The lengths of the audio segments (or ‘input lengths’) were varied throughout the evaluation process, but kept the same for each evaluation iteration. The enhancement performance was measured by means of the output signal-to-noise ratio, and registered for each evaluation iteration, along with the inference response time and memory usage through time.

The techniques that were characterized in this manner were Spectral Feature Mapping with Mimic Loss [16], MetricGAN+ [17], and Demucs-Denoiser [18]. The main motivation behind this characterization is to provide insights to the reader about what to expect from the current state-of-the-art of online speech enhancement.

This work is organized as follows. Section 2 presents a brief summary of the three chosen techniques to be characterized; Section 3 details the evaluation methodology, specifically, how the evaluation corpus was created and how the evaluation was carried out systematically; Section 4 presents the results; Section 5 provides some insights and recommendations, based on the previously presented results; and, Section 6 provides some concluding remarks and proposals for future work.

## 2. Evaluated Techniques

As mentioned beforehand, the objective of this work is to characterize the online applicability of the current state-of-the-art of speech enhancement. Recently, there have been major advances in terms of separation performance [1] and the perceptual quality of the enhanced speech [19,20]. However, these advances have been mostly carried out in offline processing scenarios, while, until very recently [16,17,18], relatively little attention has been given to online speech enhancement. It is not the objective of this work to argue why this is the case, and it would be unfair to evaluate techniques in scenarios which they were not designed for. However, this limits the amount of models that can be evaluated, even more so considering that few of them provide ways to easily replicate their results. Thus, the selection was based on the following criteria:1.Ease of replicability: the existence of a public code repository to recreate its published results.2.Online viability: it can be deduced from the original published paper that it is able to run in an online manner.3.Popularity: if a pre-trained model is publicly available in one or more speech enhancement repositories (such as Hugging Face [21]) to be downloaded and tested.

To this effect, three models were selected to be evaluated:Spectral Feature Mapping with Mimic Loss [16],MetricGAN+ [17],Demucs-Denoiser [18].

It is important to mention that this list is not to be considered exhaustive, but as a representative sample of the current state of online speech enhancement. In this section, the three techniques are described, for completeness sake.

### 2.1. Spectral Feature Mapping with Mimic Loss

The intent of the original paper [16] was not to create a speech enhancement model, but to introduce a loss function that could be used to train speech enhancement models to make Automatic Speech Recognizers (ASRs) more robust. Until that point, it was common to refine ASRs models with the output of the speech-enhancement technique since, although they were very good at removing noise and interferences, they also distorted the signal (albeit lightly) and/or inserted artifacts, both of which degraded the ASR performance. To this effect, a loss function was designed to create a training objective that estimated the target speech with high perceptual quality, not just well separated from the rest of the audio streams in the captured audio signal.

To do this, first, a spectral classifier was trained using a cross-entropy loss function [22] that aimed to classify from clean speech the presence of similar acoustic events (or ‘senones’). In parallel, a spectral mapper was trained using a mean-square error loss function (referred to as ‘fidelity loss’, $L_{f}$) that aimed to map clean speech from noisy speech, as in Equation (Equation 1).
(1)$$L_{f}\left(x,y\right)=\frac{1}{K}\sum_{k=1}^{K}{\left(y^{k}-f{\left(x\right)}^{k}\right)}^{2}$$
where *k* is the frequency bin, f(·) is the spectral mapper function, *y* is the clean signal, and *x* is the noisy input.

Then, with the weights of the spectral classifier frozen, the spectral mapper was further trained but now using the fidelity loss function in conjunction with another loss function (referred to as ‘mimic loss’, $L_{m}$), the mean square difference between the senones classified directly from a reference clean speech signal, and those that were classified from the estimated target speech. This is presented in Equation (Equation 2).
(2)$$L_{m}\left(x,y\right)=\frac{1}{D}\sum_{d=1}^{D}{\left(g{\left(y\right)}^{d}-g{\left(f\left(x\right)\right)}^{d}\right)}^{2}$$
where *D* is the dimensionality of the senone classifier, and g(·) is the D-dimensional representation of a given signal, which can be the outputs of the senone classifier either after or prior the softmax normalization.

In this regard, the fidelity loss function ($L_{f}$) represents the objective of de-noising speech, while the mimic loss ($L_{m}$) represents the objective of providing high perceptual quality of the de-noised speech. In conjunction, solving for both losses at the same time (as a simple weighted sum, as presented in Equation (Equation 3)), should result in highly de-noised, highly perceptual speech, appropriate to be used directly on pre-trained ASRs.
(3)$$L\left(x,y\right)=L_{f}\left(x,y\right)+\alpha L_{m}\left(x,y\right)$$
where α is a hyperparameter that ensures both losses are of the same magnitude.

To provide evidence of the usability of the proposed loss function, the authors of [16] trained a simple spectral mapper that consisted of a two-layer feed-forward network, with only 2048 neurons in each layer. Because of the simplicity of their proposed model, low memory requirements and small response time can be assumed.

### 2.2. MetricGAN+

From the explanation of the last technique, it could be deduced (which the authors of [16] themselves admit to) that there is a sort of inspiration of Generative Adversarial Networks (GAN) in the design of the mimic loss. This is because there is a type of “adversarial” scheme between the two trained models, from which a result is generated. This is basically the idea behind GANs [23], a ‘generator network’ that is trained and “put against” the result of an ‘adversarial network’, to generate a result. To this effect, MetricGAN [24] was introduced as a way to directly integrate into a loss function some metrics for evaluating perceptual quality, such as the perceptual evaluation of speech quality (PESQ) [25] and short-time objective intelligibility (STOI) [26]. This was proposed so that a network that uses such a loss function, would result in it mimicking the aforementioned metrics. Consequently, this was used to train the ‘discriminator network’ of the GAN; the ‘generator network’ of the GAN was trained solely relying on the adversarial loss function, as it would normally.

Later, MetricGAN+ [24] was introduced as a refinement of MetricGAN, with the following improvements:In addition to learning the metric behavior from the clean speech and estimated speech, it was also learned from the noisy speech, stabilizing the learning process of the discriminator network (which aims to mimic such metrics).Generated data from previous epochs was re-used to train the discriminator network, improving performance.Instead of using the same sigmoid function for all frequencies in the generator network, a sigmoid function was learned for each frequency, providing a closer match between the sum of the clean and noise magnitudes, and the magnitude of the noisy speech.

The authors of [24] trained a two-layered bi-directional long-short-term memory (Bi-LSTM) network [27], with only 200 neurons in each layer, followed by two fully connected layers, a rectified linear unit (ReLU) layer [28] and the learned sigmoid layer. The size of the network is bigger than the one trained form SFM-Mimic, but it is still small, from which the same assumption of memory requirements and response times can be made.

### 2.3. Demucs-Denoiser

A sizable part of the speech-enhancement techniques that are currently in the literature aim to solve the problem by estimating a frequency mask that, when applied to the input signal, results in an enhanced output [2]. This implies that there is a considerable part of the training process that needs to be focused in classifying each time-frequency bin as either part of the target speech or not. This may not only require a high amount of memory resources, but the transformation to and from the frequency domain may add on to the response time of the model. To this effect, the Demucs model [29] was introduced first to separate music instruments from musical recordings, directly in the time domain. The Demucs-Denoiser model [18] is an evolution of such model focused on speech enhancement, which from a musical point of view, aims to separate one “instrument” (target speech) from the rest of the “recording” (noisy speech).

The Demucs model network employs a type of encoder–decoder architecture, with skip connections between symmetric parts of the encoding and decoding parts of the model network. This structure is also known as a U-Net structure, which was first used for biomedical imaging [30]. Once the audio data are encoded, causal prediction is carried out to de-noise it at an encoded level. Then, these de-noised encoded data are decoded (with the help of the skip connections) to produce the estimated target speech waveform. The authors of [18] also employed an up-sampling scheme, with which the encoding–decoding process provided good results. Additionally, for training purposes, two loss functions were used that compared the clean and estimated target speech, an L1 loss function over the waveform domain, and a short-time Fourier transform (STFT) loss over the time-frequency domain.

The encoding part of the model was conformed by five meta-layers. Each, in turn, was conformed by a 1-dimensional convolutional layer with a rectified linear unit (ReLU) layer [28], and another convolutional layer with a gated recurrent unit (GRU) layer [31]. The decoder part was made up of the same meta-layers but in reverse, each with its respective skip connection. The causal prediction was carried out by a 2-layer bi-directional long-short-term memory (LSTM) network [27]. The resulting model may appear to require a relatively large amount of memory (as opposed to the other two techniques) which, in turn, may imply large response times. However, the authors of [18] verified that it was able to run in an online manner, although the input lengths were not as varied as it is presented in this work.

The model was trained with different training datasets, producing different versions of their models. The model characterized in this work is the one trained with the DNS dataset [32], with 64 initial output convolutional channels.

## 3. Evaluation Methodology

The objective of this work is to characterize speech-enhancement techniques under different conditions that are applicable to online processing scenarios. Thus, in addition to evaluating techniques in the presence of different levels of noise, different levels of reverberation, and different amounts of speech interferences, the evaluation is to be carried out using different input lengths. To this effect, an evaluation corpus was synthesized for this purpose, and a windowing methodology was implemented that simulates what these techniques should be put through when carrying out speech enhancement in an online manner. Finally, when evaluating the techniques, several metrics are measured that are relevant to the quality of the result (mainly signal-to-interference ratio), as well as to the implementation viability (such as response time and memory consumption).

### 3.1. Input Lengths and Online Processing

It is crucial to subject the speech-enhancement techniques to different input lengths ($N_{w}$) in a systematic and repeatable manner, so that the comparisons between techniques are “apples-to-apples”. Thus, a windowing methodology was implemented that, given an specific input length and a given recording (the manner in which it is synthesized is detailed in Section 3.2) divides such a recording into subsequent audio segments of that length, as presented in Equation (Equation 4).
(4)$$x_{t}=x\left[\left(t*N_{w}\right):\left(\left(t+1\right)*N_{w}\right)-1\right]$$
where $x_{t}$ is the *t*th audio segment of *x*, and $N_{w}$ is the given input length. The last segment is zero-padded if its length is smaller than the given input length.

Each audio segment is then fed to the speech-enhancement technique, chronologically. The audio segments that the technique outputs are concatenated, in the sequence they were fed. Finally, the result of this concatenation is sliced so that it is the same length of the given recording, removing the zeroes that were padded to the last segment, providing the final estimated output. From this output, the performance metrics are calculated (detailed in Section 3.4.1).

### 3.2. Corpus Synthesis, Evaluated Variables

As mentioned beforehand, a corpus is required to carry out the characterization of the speech-enhancement techniques. To this effect, the ‘dev-clean’ subset of the LibriSpeech corpus [15] was used as the basis of this corpus. The LibriSpeech corpus is a set of recordings of users reading novels, as part of the LibriVox project [33]. It bears several sub-sets of recordings, aimed to be used for training and validating speech-related applications. The ‘dev-clean’ subset is used for validation, and it has little to no noise or perturbations, which makes them an appropriate point of reference from which the corpus can be based on.

The following perturbations were added to synthesize the evaluation corpus, each with its own different varying scales:Reverberation,Signal-to-noise ratio,Number of interferences,Input signal-to-interference ratio.

These perturbations were each added sequentially to various recordings of the ‘dev-clean’ subset of LibriSpeech to synthesize recordings, which, in turn, conform the evaluation corpus. The order in which these perturbations were added is (1) a target speech source is randomly chosen from LibriSpeech; (2) if need be, several interferences are also randomly chosen, and scaled to a given input SIR; (3) reverberation is added to all sound sources; (4) all sources are mixed; and, finally, (5) noise is added to the mixture with a given SNR.

These perturbations were added using their respective implementations inside the Speechbrain framework [34]. The Speechbrain implementation of the aforementioned perturbations have been empirically reviewed and used by a considerable part of the speech enhancement community, which provides certainty of their efficacy. Although these implementations are mainly used to create the training data on the fly when using Speechbrain to train speech enhancement models, for the purposes of this work, they were used to perturb offline recordings that emulate speech in real-life scenarios.

#### 3.2.1. Reverberation Scale

The *AddReverb* function of the Speechbrain framework was used to add a reverberation effect to the recording. This implementation requires a set of room impulse responses (RIRs). The Speechbrain framework automatically uses the RIRs described in [35]. This set of RIRs is conformed from various sub-sets, and in this work the ‘simulated large room’ sub-set was used. It was chosen because it was the one that provided a more perceptual sense of reverberation from several subjective listening sessions.

Additionally, the *AddReverb* function can “scale” the amount of reverberation added by providing a scale factor (*rir_scale_factor*), which is a number from 0 and up. The 0 represents no reverberation added; 1, one scale of reverberation; 2, two scales; etc. From subjective listening sessions, a scale factor of 3 was considered to be a “high level of reverberation”, and established to be the high limit of reverberation added. This scale factor is referred here as ‘reverberation scale’.

#### 3.2.2. Signal-to-Noise Ratio

The *AddNoise* function of the Speechbrain framework was used to add white noise to the recording. Although this function can add different types of noises (given a set of recordings of such noises), it was decided to only add white noise since the amount of energy can be easily controlled. In this regard, the *AddNoise* function receives, as an argument, the signal-to-noise ratio (SNR) of the added noise. However, this function receives a lower and upper limit, and adds it in a random manner.

To have better control of how much noise is being added, both the lower and upper SNR limit were set to the same value. This SNR value was progressively increased from an established lower limit to an established upper limit. In this manner, the amount of recordings with the same SNR value will be the same. The established lower limit was set at −10 dB, which represents the very challenging scenario of having more noise than clean speech. The established upper limit was set at 30 dB, which represents the trivial scenario of imperceptible noise.

#### 3.2.3. Number of Interferences

The *AddBabble* function of the Speechbrain framework was used to add interferences (other speakers) to the recording as additional audio streams (also known as ‘babble’). The interferences were recordings of other speakers from the same dev-clean subset of LibriSpeech. To have better control on what recordings have babble added onto them, the probability of a recording having babble (*mix_prob*) was set to 100% if such a recording was meant to have babble added; if not, the *AddBabble* function was not run for such a recording. The lower limit of the number of interferences was set to 0 and 3 was the upper limit.

#### 3.2.4. Input Signal-to-Interference Ratio

In addition to adding interferences (or ‘babble’), the amount of babble energy (here referred to as ‘input SIR’) was also controlled as an argument of the *AddBabble* function of the Speechbrain framework. The same method of establishing lower and upper limits of SNR values described Section 3.2.2 was used to establish the input SIR limits.

To avoid audio clipping, the *AddBabble* function applies the SIR value using the energy of the whole sum of interference streams, not the energy of each interference stream. This results in a small “perceived” increase in SIR the more interferences are added, since the energy of each interference stream is somewhat “diluted” by the number of interferences.

### 3.3. Evaluation Corpus

Table 1 presents all the evaluated variables and all their specific values.

Each combination of values in Table 1 is referred to here as an ‘evaluation configuration’. Considering all the different values, this resulted in 11,664 possible evaluation configurations. However, in terms of synthesized recordings, the input length is varied during the evaluation, and not part of the perturbations that were applied to synthesize recordings. So this resulted in 1296 ‘perturbation configurations’.

The ‘dev-clean’ subset of the LibriSpeech corpus is conformed by 40 speakers, with between 20 and 50 recordings for each speaker. From all these clean recordings, 100 were randomly selected and all the perturbation configurations were applied to them. This resulted in 129,600 synthesized recordings with which the techniques are to be evaluated. This means that for every evaluation configuration, taken from the combination of evaluated variable values in Table 1, there are 100 different distinct recordings from which evaluation metrics are to be calculated. For repeatibility, the scripts used to synthesize these recordings are freely accessible at https://github.com/balkce/seeval (accessed on 27 April 2023).

All recordings are sampled at a sampling frequency ($f_{s}$) of 16 kHz. This was inherited from the LibriSpeech corpus, but, coincidentally, it is the same sample rate with which all three speech-enhancement techniques were trained.

Finally, for each synthesized recording, an accompanying text file was created to register the file path of the recording of the clean target speech, as well as the file paths of the recordings of all the interferences that are part of the added babble.

### 3.4. Evaluation Metrics

As explained in the last section, all three speech-enhancement techniques were applied to each evaluation configuration, with 100 different recordings. This resulted in having 100 results per evaluation configuration, from which the following three metrics were recorded:Output signal-to-interference ratio,Inference response time and real-time factor,Memory usage.

For repeatability, the scripts used to run the whole evaluation and record all the results are freely accessible at https://github.com/balkce/seeval (accessed on 27 April 2023).

#### 3.4.1. Output Signal-to-Interference Ratio

The output signal-to-interference ratio (SIR) can be calculated as presented in Equation (Equation 5).
(5)$$SIR=\frac{\left|\right|s_{T}{\left|\right|}^{2}}{\sum_{i}^{I}\left|\right|s_{i}{\left|\right|}^{2}},$$
where ${\left||.|\right|}^{2}$ is the norm operation used to calculate the energy from a given signal, $s_{T}$ is the target signal, $s_{i}$ is the *i*th interference, and *I* is the number of interferences.

As it can be deduced, to synthesize a recording with a given SIR is trivial, since the target signal and the interferences are separated from the start. However, calculating the output SIR is not as trivial, since the energy of the clean target speech and of the interferences need to extracted from the estimated target output.

A popular method, known as *bss_eval_sources* [36], is usually used to calculate the output SIR. It requires to be fed all the clean versions of the speech signals inside the mixture, including the interferences, which are available in the accompanying text files for each recording. Unfortunately, it also requires to be fed all their estimated versions as well. Because of the nature of speech enhancement, only the estimated target speech is provided; the interferences are not estimated. Fortunately, *bss_eval_sources* calculates another metric, referred to as ‘signal-to-distortion ratio’ (SDR), and calculates it such that (in an admittedly over-simplified manner) anything that is not the target signal is considered as a “distortion”. Thus, if *bss_eval_sources* is fed, only the clean target speech and the estimated target speech, the SDR is equivalent to the SIR presented in Equation (Equation 5). The implementation of *bss_eval_sources* that was used is part of the *mir_eval* package [37].

An output SIR of 20 dB can be considered of “high” enhancement performance.

#### 3.4.2. Inference Response Time and Real-Time Factor

The response time was measured just for the inference part of the speech-enhancement technique (no re-sampling or file loading was included as part of the response time). Because the synthesized recording is split into several audio segments, each of a given input length ($N_{w}$) as part of the evaluation configuration, several inferences are carried out for each evaluation configuration (one for each audio segment). Thus, the recorded response time is the average of the inference response times of all the fed audio segments.

It is important to mention that response times are highly dependent on the specifications of the machine that is used to carry out the evaluation. In this case, the evaluation machine had an AMD Ryzen 9 3900X 12-Core Processor; no other processing unit (such as a GPU with CUDA) was used as part of the inference of the speech-enhancement techniques. The reason why this was chosen was because it was of interest to see how well will the techniques respond without a GPU to handle the inference, but with an above-average CPU. This scenario is one that an enthusiast user may find themselves in, and establishes a base of reference for the response time that they can expect in their own computers.

Additionally, the real-time factor (RTF) was calculated and recorded. The RTF measures how viable a signal processing technique is to be able to run in an online manner, and is calculated as shown in Equation (Equation 6).
(6)$$RTF=\frac{\tau_{p}+\tau_{a}}{N_{w}}*f_{s}$$
where $N_{w}$ is the input length (in number of samples), $f_{s}$ is the sampling rate; $\tau_{p}$, the inference response time (which was recorded, in seconds); and $\tau_{a}$ is the sum of the response times (in seconds) for pre-processing and post-processing the audio segment. The latter was measured through a series of independent tests, focusing on re-sampling from the original audio sampling rate to the sampling rate of the recordings the techniques were trained with (16 kHz), and vice versa.

If RTF>1, it means that the technique is not viable to be run online and ‘overruns’ are to be expected. When an overrun occurs, since the audio segment was not processed in time, it is discarded and usually replaced with zeroes in the concatenation, which results in heavy distortions in the final estimation, or even in complete silence if RTF≫1. On the other hand, online viability can be implied if RTF<1. However, if RTF≲1, one can still expect overruns to occur, but in a more sporadic fashion (since response times vary from audio segment to audio segment), thus, heavy distortions in the final estimation are to be expected. Thus, an RTF of 0.5 is usually considered preferable.

#### 3.4.3. Memory Usage

The amount of memory the technique occupied in RAM (in bytes), at the end of the set of inferences for each evaluation configuration, was recorded. This was obtained through the assessment of the “Resident Set Size” field of the built-in memory management system in Debian Linux, reported by the *psutil* package.

## 4. Results

The impact of each evaluated variable is to be inspected, in respect to the input length evaluated variable. This means that most of the figures in this section have, as the horizontal axis, the varying input lengths.

For full transparency, the whole set of results are provided as Appendix A to this work. However, for simplicity, when assessing each variable, the other variables are set to their default value shown in Table 2, unless specified otherwise.

Additionally, since 100 different recordings were synthesized for every evaluation configuration, the figures are shown as box plots, so that the distribution of the results are shown for each evaluation configuration, along with its median value.

The “box” in the box plot represents the two middle quartiles of all the measured SIR results, and the line that crosses the middle of the box is the median of all the measured SIR results. The difference of the minimum value and maximum value inside this box is known as the inter-quartile range (IQR). To this effect, the box also has two “whiskers”; the bottom of the lower whisker represents the value that is 1.5∗IQR below the minimum value of the box; the top of the upper whisker represents the value that is 1.5∗IQR above the maximum value of the box. Finally, ‘outliers’ are values that are outside the range between the bottom of the lower whisker and the top of the upper whisker, and are plotted as points.

In the following sub-sections, the performance results for each evaluated variable are presented, and in Section 5 the insights from these results are discussed, with some recommendations for future work.

### 4.1. Output SIR vs. Input Length

In Figure 1, the separation performance (SIR) of all three techniques are shown, in respect with the input length.

As can be seen, there is a clear upper trend of performance when increasing the input length for all three techniques. However, it can be seen that Demucs-Denoiser is more robust against this change, having the best performance with small input lengths (<4096 samples, 0.256 s) in both terms of median SIR and SIR variance (presenting a more predictable behavior). On the other hand, MetricGan+ is outperformed by the other two techniques with every input length.

### 4.2. Output SIR vs. Reverberation Scale

In Figure 2, Figure 3 and Figure 4, the impact of reverberation across different input lengths are shown for MetricGan+, SFM-Mimic and Demucs-Denoiser, respectively.

As can be seen, all models are impacted by reverberation. The input length does not seem to provide any considerable effect, except with MetricGan+ that does appear to benefit from longer input lengths (>32,768 samples, 2.048 s).

### 4.3. Output SIR vs. Number of Interferences

In Figure 5, Figure 6 and Figure 7, the impact of the number of interferences across different input lengths are shown for MetricGan+, SFM-Mimic, and Demucs-Denoiser, respectively. The input SIR was set to 0 dB for these results.

As can be seen, all models are severely impacted by having interferences, with a drop of more than 10 dB in all circumstances; Demucs-Denoiser also presents a higher SIR variance, resulting in more unpredictable behavior. Additionally, a slight upper trend can be seen when increasing the number of interferences; this could be explained by the slight increase in input SIR detailed in Section 3.2.4.

### 4.4. Output SIR vs. Input SIR

In Figure 8, Figure 9 and Figure 10, the impact of the input SIR across different input lengths are shown for MetricGan+, SFM-Mimic, and Demucs-Denoiser, respectively. The number of interferences was set to 1 for these results.

As can be seen, all models benefit greatly by increasing the input SIR. It is important to note that with an input SIR between −5 and 5 dB, MetricGAN+ and Demucs-Denoiser provide a large variance in their performance. This may indicate that the techniques are randomly “choosing” to estimate either the interference or the target speech when they have similar energies. This will be discussed further in Section 5.

### 4.5. Output SIR vs. SNR

In Figure 11, Figure 12 and Figure 13, the impact of the SNR across different input lengths are shown for MetricGan+, SFM-Mimic, and Demucs-Denoiser, respectively. It is important to note that a SNR of 30 dB is virtually imperceptible.

As it can be seen, all models benefit greatly by increasing the SNR. It is important to note that although Demucs-Denoiser outperforms the other two in terms of median output SIR, it still provides a similar performance as the other two techniques in the cases of very low SNR (−10 dB) and very small input lengths (<2048 samples, 0.128 s).

### 4.6. Real-Time Factor

The response times ($\tau_{p}$) of all three techniques across different input lengths are shown in Figure 14. No response times were measured when feeding the whole recording to the techniques.

As detailed in Section 3.4.2, the real-time factor is calculated as $RTF=\frac{\tau_{p}+\tau_{a}}{\tau_{c}}$, where $\tau_{c}$ is the input length (in seconds) and $\tau_{a}$ is the sum of the response times of pre-processing and post-processing the audio segment. These were measured independently, focusing on re-sampling to and from the audio segment sampling rate, since this is the main pre- and post-processing step that is carried out during online speech enhancement. A $\tau_{a}$ was obtained per input length averaging the response time of 100 iterations of pre- and post-processing an audio segment of such length. The real-time factors of all three techniques across different input lengths are shown in Figure 15.

As can be seen, all three models decrease their real-time factors (RTFs) the longer the input lengths, even when their inference response time increases. This implies that the increase in inference response time is considerably slower than the increase in input length. As a result, their online viability increases the more information is provided, which may seem counter-intuitive, since there are more data to process; however, this also means that there is more time to process it, at a price of added latency. Furthermore, there seems to be plateau of the RTF value around the input length of 8192 samples (0.512 s) where more information does not seem to decrease the RTFs considerably in all three techniques, with no clear impact on the variance.

Additionally, MetricGan+ provides the lowest real-time factors across all circumstances, pointing to it being the most viable to be run in an online manner. This is important since, as it can be seen in the previous sections, it usually provides the lowest output SIR of all three techniques. This hints that the enhancing performance of MetricGan+ was, in a way, traded off by its quick response times.

To be fair, however, it is important to note that all three techniques have real-time factors well below its upper limit of online viability (RTF=1). Meaning, that all three are viable to be run in an online manner.

### 4.7. Memory Usage

The memory usage (in megabytes) through the first 100 evaluation iterations of all three techniques can be seen in Figure 16. As can be seen, MetricGAN+ starts with the lowest memory usage in those first 100 evaluation iterations.

However, an extension to 10,000 evaluation iterations is shown in Figure 17, and it can be seen that Demucs-Denoiser provides a more predictable memory usage through time. Meaning, even if it does require considerable more memory to start out than MetricGAN+, it may be preferable to it since MetricGAN+ (along with SFM-Mimic) will start increasing their memory usage substantially.

In any case, it is important to note that, even in the best case scenario, all three models require more than 500 MB of memory to carry out speech enhancement. In the worst case scenario, the memory usage for MetricGAN+ and SFM-Mimic requires to unload the model from memory and reloaded again so that memory usage stays within a practical range.

## 5. Discussion

As expected, increasing SNR and/or input SIR and decreasing the reverberation, increases the performance across all input lengths. It could be argued that with small input lengths (<4096 samples, 0.256 s), the performance does not seem to be as impacted by the SNR, input SIR, and reverberation as with longer input lengths. However, the performance in these cases is already low to begin with. Thus, there seems to be an output SIR floor for MetricGAN+ and SFM-Mimic, disturbing the signal further (with interferences, noise and reverberation) will not provide worse performance.

The presence of interferences (even just one) impacts all three techniques, regardless of the input length. Demucs-Denoiser, which performs well even with the smallest input length, seems to be the least robust when adding interferences, to the point of performing similarly to the other two techniques. This is worth noticing since Demucs-Denoiser usually outperforms the other two techniques in similar input length scenarios. It is important to remember that speech-enhancement techniques aim to “separate” the target speech stream from the audio signal, assuming that only one speech stream is present in the audio signal. If this is not the case, the techniques will either (a) implicitly “decide” which speech stream to focus on based on unforeseen biases in the training data, usually separating the loudest one; or (b) mix all of the speech streams into one stream that is separate from the rest of the audio signal. This explains the severe impact on the performance of all techniques when adding interferences.

Furthermore, there is an increase in variance in performance when increasing the SNR for MetricGAN+ and for SFM-Mimic, for all input lengths. The SNR values where this phenomenon occurs are moderately high (>10 dB), which indicates that these techniques, when they fail, appear to do so in a more predictable manner than when they succeed. This is actually a good thing since it provides a clear point from which to refine them. As for Demucs-Denoiser, although this SNR-to-performance-variance is still present, it is at a much lower rate, which demonstrates that it is much more predictable (when both failing and succeeding) across the SNR value range.

There is also an increase in performance variance when increasing the SIR for MetricGAN+ with all input lengths and for SFM-Mimic with input lengths smaller than 32,768 samples (2.048 s). Again, the SIR values where this phenomenon occurs are moderately high (>10 dB), which reinforces the intuition provided earlier, although its appearance is more subtle. Interestingly, MetricGAN+ and Demucs-Denoiser tend to have a high performance variance when input SIR values are between −5 and 5 dB, meaning, when the target speech and the interferences have similar energies. As mentioned before, this may indicate that both techniques are “choosing” randomly which speech stream to separate (either the target or interference), hinting that some sort of selection mechanism may be needed for these scenarios.

One could also argue that MetricGAN+ should be avoided, considering that it is consistently outperformed by the other two techniques, and that it is the most impacted by small input lengths (<4096 samples, 0.256 s). However, as mentioned earlier, it provides the best real-time factor of all three, which may hint that such a dip in performance was at the cost of its small response times. This is not to say that MetricGAN+ is “efficient”, since it (along with SFM-Mimic) have a clear problem with their memory usage through time. The author of this work went to great lengths to ensure that there were not any memory leaks in the scripts used in the evaluation process. This is further evidenced by the fact that the Demucs-Denoiser did not suffer from this upward trending memory consumption, while using the same evaluation scripts. Thus, it can be concluded that both MetricGAN+ and SFM-Mimic are in dire need of memory usage tuning.

All three techniques are viable to be run in an online manner, providing better results (both in terms of output SIR and RTF) the greater the input lengths. However, it is important to note that increasing the input length will result in added latency for the user. To this effect, the author of this work recommends an input length between 4096 and 8192 samples (between 0.256 s and 0.512 s), for SFM-Mimic and Demucs-Denoiser. The reasoning behind this recommendation is that the impact of input SIR, SNR and reverberation in separation performance seem to plateau around these lengths, while providing an RTF value that is close to their respective lowest, along with a latency that could be tolerated by the average user.

## 6. Conclusions

Speech enhancement performance has been greatly increased by the employment of deep learning-based techniques. Carrying out these techniques in an online manner is of great interest, but few systematic evaluations for these scenarios have been carried out. In this work, a thorough characterization of three popular speech-enhancement techniques (MetricGAN+, SFM-Mimic, and Demucs-Denoiser) was carried out by emulating an online audio processing scheme. It involved sequentially feeding audio segments (of varying input lengths) of synthesized recordings to the techniques, and concatenating the results. Synthesized recordings were created from 100 randomly selected recordings of the ‘dev-clean’ subset of the LibriSpeech corpus, which were then disturbed by adding varying amounts of noise, varying amounts of interferences with varying amounts of energy, and varying amounts of reverberation, resulting in a sizable evaluation corpus. The scripts to create this evaluation corpus, as well as to carry out the systematic characterization, are freely accessible at https://github.com/balkce/seeval (accessed on 27 April 2023).

It was found that small window lengths (<4096 samples, 0.256 s) are detrimental to the enhancement performance (measured as output SIR). However, Demucs-Denoiser was the least affected by it. Furthermore, this technique provided the least result variance when increasing SNR, providing predictable results, although it was affected overall similarly as SFM-Mimic. On the other hand, MetricGAN+ was consistently outperformed by the other two techniques, but it was observed that this could be considered as a trade-off for its low response times. However, all three techniques provided real-time factor values well below the threshold for online viability, with the author of this work recommending lengths between 4096 and 8192 samples (between 0.256 s and 0.512 s) for SFM-Mimic and Demucs-Denoiser, since they provide a balance between output SIR (against all disturbances), RTF, and latency.

All three models were severely impacted by reverberation, for all input lengths. They were also severely impacted by the presence of interferences; with even just one was enough to see dips in performance of more than 10 dB. In terms of SIR, although they were all impacted as expected, Demucs-Denoiser showed an increase in result variance when the target speech and the interference had a similar energy (input SIR between −5 and 5 dB). This hints at the possibility of “confusion” of which speech stream to “focus on”, resulting in the techniques randomly choosing to separate the interference instead of the target speech. It is proposed, as future work, to implement a type of selection mechanism for the Demucs-Denoiser so that this issue is resolved.

## Figures and Tables

**Figure 1 sensors-23-04394-f001:**
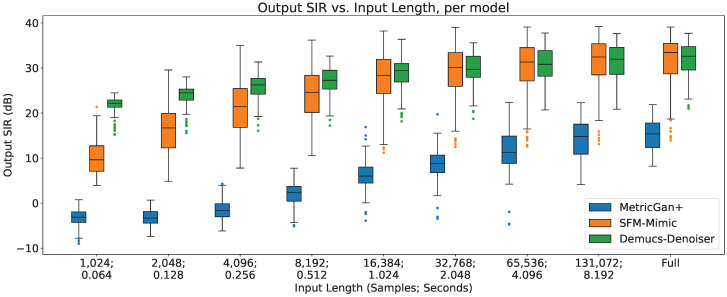
Output SIR vs. input length, per model.

**Figure 2 sensors-23-04394-f002:**
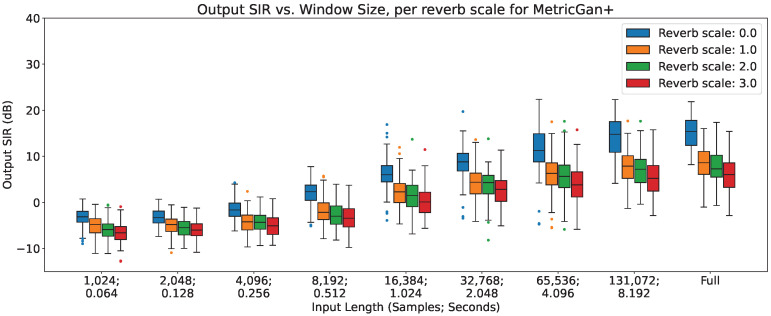
Output SIR vs. input length, per reverberation scale, for MetricGan+.

**Figure 3 sensors-23-04394-f003:**
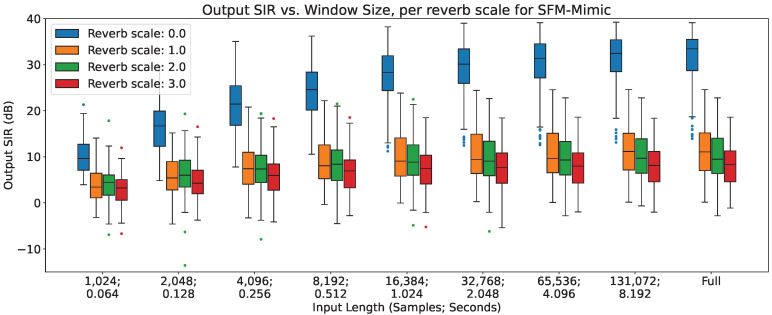
Output SIR vs. input length, per reverberation scale, for SFM-Mimic.

**Figure 4 sensors-23-04394-f004:**
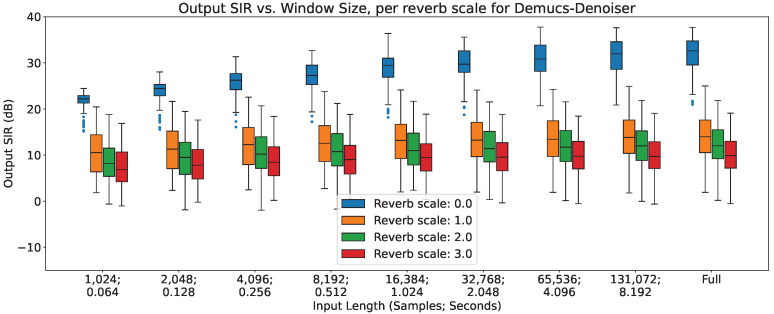
Output SIR vs. input length, per reverberation scale, for Demucs-Denoiser.

**Figure 5 sensors-23-04394-f005:**
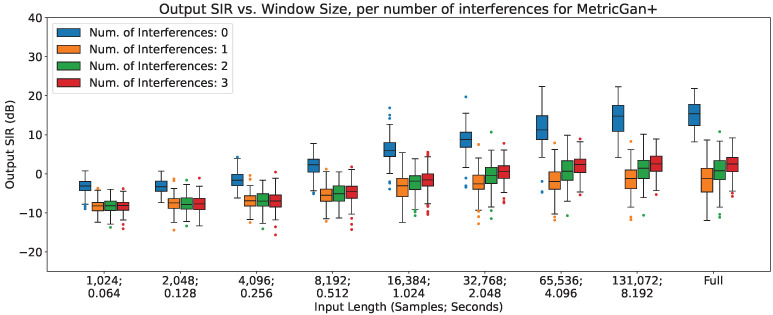
Output SIR vs. input length, per number of interferences, for MetricGan+.

**Figure 6 sensors-23-04394-f006:**
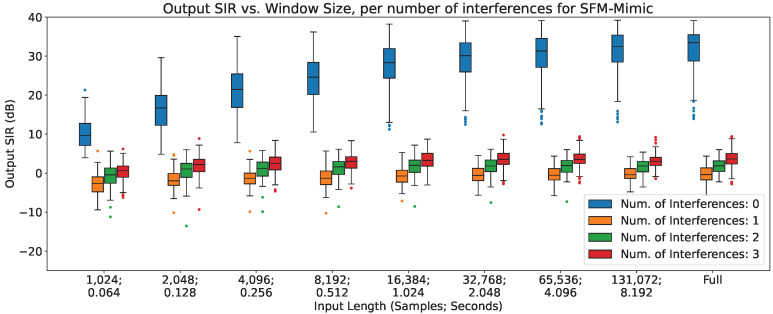
Output SIR vs. input length, per number of interferences, for SFM-Mimic.

**Figure 7 sensors-23-04394-f007:**
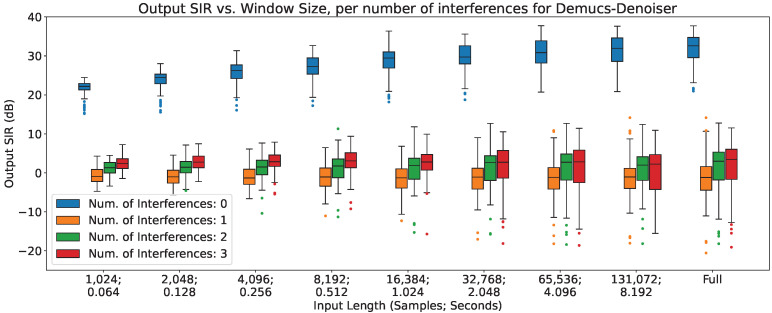
Output SIR vs. input length, per number of interferences, for Demucs-Denoiser.

**Figure 8 sensors-23-04394-f008:**
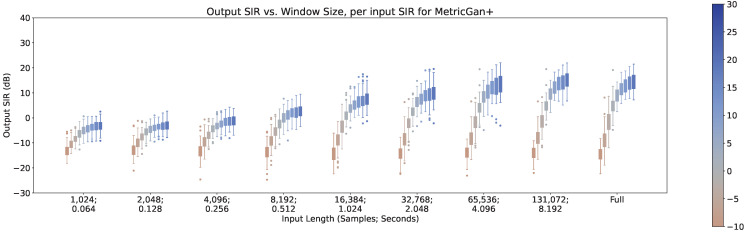
Output SIR vs. input length, per input SIR, for MetricGan+.

**Figure 9 sensors-23-04394-f009:**
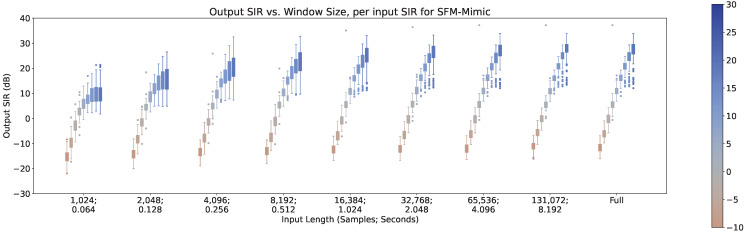
Output SIR vs. input length, per input SIR, for SFM-Mimic.

**Figure 10 sensors-23-04394-f010:**
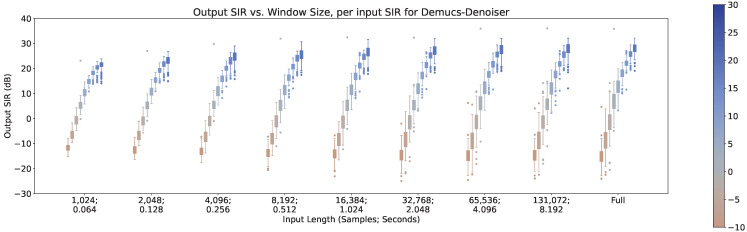
Output SIR vs. input length, per input SIR, for Demucs-Denoiser.

**Figure 11 sensors-23-04394-f011:**
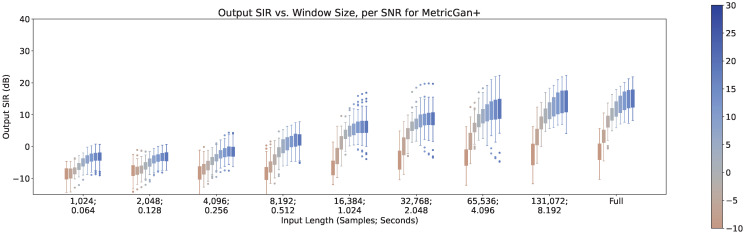
Output SIR vs. input length, per SNR, for MetricGan+.

**Figure 12 sensors-23-04394-f012:**
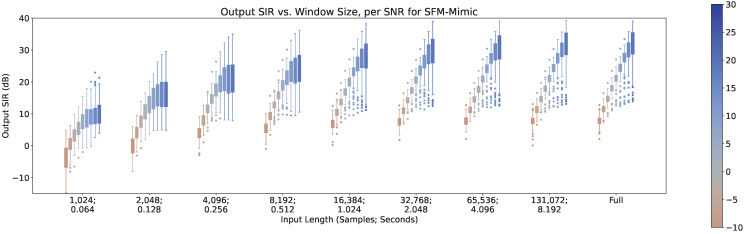
Output SIR vs. input length, per SNR, for SFM-Mimic.

**Figure 13 sensors-23-04394-f013:**
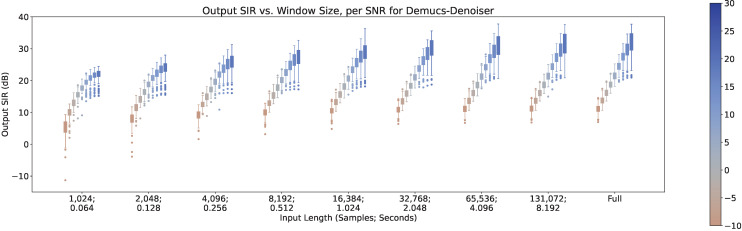
Output SIR vs. input length, per SNR, for Demucs-Denoiser.

**Figure 14 sensors-23-04394-f014:**
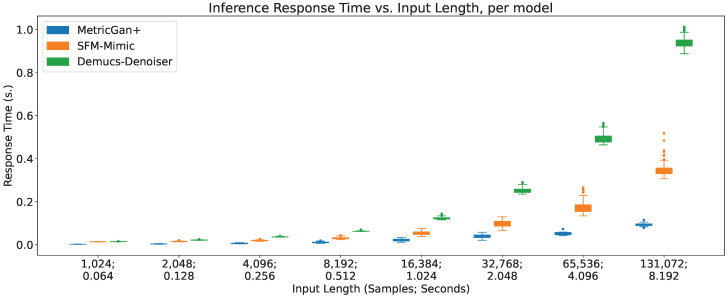
Response time vs. input length, per model.

**Figure 15 sensors-23-04394-f015:**
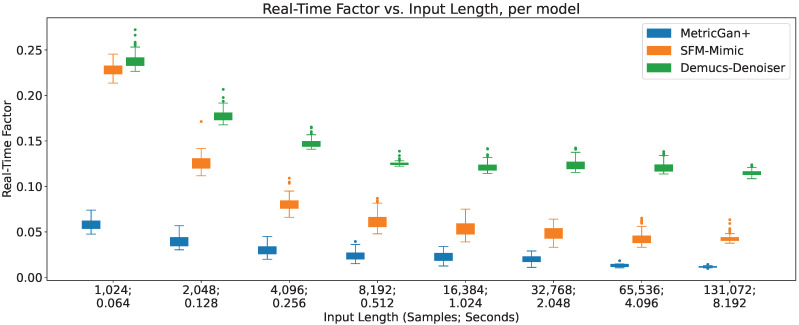
Real-time factor vs. input length, per model.

**Figure 16 sensors-23-04394-f016:**
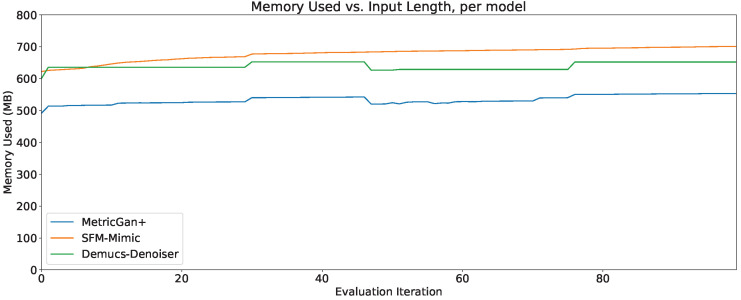
Memory usage through 100 evaluation iterations, per model.

**Figure 17 sensors-23-04394-f017:**
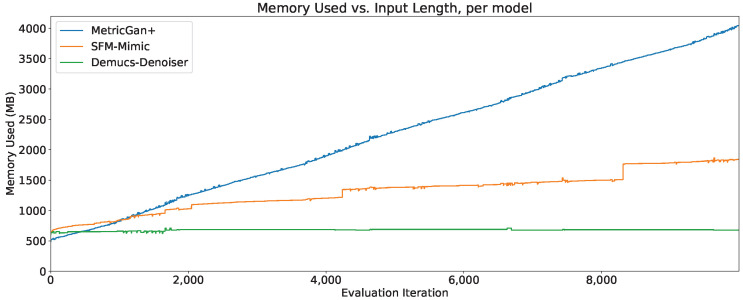
Memory usage through 10,000 evaluation iterations, per model.

**Table 1 sensors-23-04394-t001:** Evaluated variables and their values.

Evaluated Variable	Values
Input Length (samples)	1024, 2048, 4096, 8192, 16,384, 32,768, 65,536, 131,072, Full recording
Reverberation Scale	0, 1, 2, 3
Signal-to-Noise Ratio (dB)	−10,−5, 0, 5, 10, 15, 20, 25, 30
Number of Interferences	0, 1, 2, 3
Input Signal-to-Interference Ratio (dB)	−10,−5, 0, 5, 10, 15, 20, 25, 30, 100 ^†^

^†^ When the number of interferences is 0, the Input SIR is set to 100, but only symbolically, since it is not used.

**Table 2 sensors-23-04394-t002:** The default values for all evaluated variables, except input length.

Variable	Default Value
Reverberation Scale	0
Signal-to-Noise Ratio (dB)	30
Number of Interferences	0
Input Signal-to-Interference Ratio (dB)	100 ^†^

^†^ When the number of interferences is 0, the Input SIR is set to 100, but only symbolically, since it is not used.

## Data Availability

All of the scripts employed to carry out the characterization presented in this work are publicly accessible in https://github.com/balkce/seeval (accessed on 27 April 2023).

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
