# Peer review of "Characterization of Deep Learning-Based Speech-Enhancement Techniques in Online Audio Processing Applications"

_sensors, 2023, doi:10.3390/s23094394_

Round 1
Reviewer 1 Report
This article evaluates the importance of three speech enhancement techniques in which characterisation is carried out using a systematic evaluation protocol based- Speechbrain framework. However, authors did a great work and extensive research for highlighting the importance of applications that process audio online.
I would like to thank the authors for their efforts and I would suggest that this manuscript is ready for publication without further revision.
Author Response
The author greatly appreciates the response from the reviewer. It appears that no further response is necessary.
Reviewer 2 Report
Thank you for your hard work. The presented paper is well-written, and the article's subject is evident. However, before publication, it is necessary to correct some mistakes.
Line Things to improve
20 There is an "in real-time scenario." It has to be corrected to "in a real-time scenario."
107-119 The described parameters should also be written as the equations.
147, 170, 171, 180, 181 This paper has only one author. However, in the sentences, instead of "author," is the word "authors."
175, 175, 178 The abbreviations: ReLU, GRU, LSTM are written without explanation or reference.
191 The "windowing methodology" has no explanation or bibliography either.
224-226 The sentence "These implementation ... " has no reference.
316 The symbol "|" has to be changed to "or."
342 It is necessary to describe Nw and fs in Equation 2.
573 The reference is written using only lowercase. Therefore, it has to be corrected.
The publication year must be written in bold in the following references: 6, 10, 15, 16, 17, 18, 21, 23, 25, 27, 29, 30, and 32.
Author Response
The author greatly appreciates the comments from the reviewer. Here is a point-by-point response to the reviewer's comments:
- There is an "in real-time scenario." It has to be corrected to "in a real-time scenario."
>>> Fixed.
- The described parameters should also be written as the equations.
>>> A reference was added for cross-entropy loss, while the other three loss functions are now described with equations.
- This paper has only one author. However, in the sentences, instead of "author," is the word "authors."
>>> These sentences refer to the work of the authors of the original papers. This was clarified throughout the manuscript.
- The abbreviations: ReLU, GRU, LSTM are written without explanation or reference.
>>> Fixed, and references were added.
- The "windowing methodology" has no explanation or bibliography either.
>>> A brief explanation (with an accompanying equation) was added at the start of section 3.1.
- The sentence "These implementation ... " has no reference.
>>> This sentence has been changed to "The Speechbrain implementation of the aforementioned perturbations" which is clearer.
- The symbol "|" has to be changed to "or."
>>> It was a typo. It has been fixed.
- It is necessary to describe Nw and fs in Equation 2.
>>> These variables are now part of the aforementioned equation, and are now described in the text.
- The reference is written using only lowercase. Therefore, it has to be corrected.
>>> Fixed.
- The publication year must be written in bold in the following references: 6, 10, 15, 16, 17, 18, 21, 23, 25, 27, 29, 30, and 32.
>>> The year field for proceeding-type references is not set in boldface by the default MDPI reference template.
Reviewer 3 Report
The whole paper is well organised and clearly written.
Scientific contribution is not large (there is no new methods), but huge effort has been devoted to compare online performance of selected speech enhancement techniques.
221: “These perturbations were each added sequentially to various recordings…” – In which order?
The chapter “Discussion” is not so clear as other parts of text.
The list of references is good but could be improved.
All Figures (1-7, particularly 8-10, but also 11-13) show “Output SIR...” – not “SIR...” (since there is “input SIR”). Please, write in their legends “Output SIR…”.
The paper has one author, while use plural “the authors” at several places in the text.
Minor typos:
“where…” after equations (1) and (2) should not be indented like regular paragraphs.
195: .) - ).
339: was calculated was recorded (?)
346: (16 kHz.), - (16 kHz),
Author Response
The author greatly appreciates the comments from the reviewer. Here is a point-by-point response to the reviewer's comments:
- "These perturbations were each added sequentially to various recordings" – In which order?
>>> The order in which these perturbations were added is: 1) a target speech source is randomly chosen from LibriSpeech; 2) if need be, several interferences are also randomly chosen, and scaled to a given input SIR; 3) reverberation is added to all sound sources; 4) all sources are mixed; and, finally, 5) noise is added to the mixture with a given SNR. This text was added to the manuscript following the aforementioned sentence.
- The chapter "Discussion" is not so clear as other parts of text.
>>> Several sentences and phrases were improved in this section, so as to make it as clear as other parts of the manuscript.
- The list of references is good but could be improved.
>>> The following references were added:
. Gordon-Rodriguez, E.; Loaiza-Ganem, G.; Pleiss, G.; Cunningham, J.P. Uses and abuses of the cross-entropy loss: Case studies in modern deep learning 2020.
. Hara, K.; Saito, D.; Shouno, H. Analysis of function of rectified linear unit used in deep learning. In Proceedings of the 2015 international joint conference on neural networks (IJCNN). IEEE, 2015, pp. 1–8.
. Dey, R.; Salem, F.M. Gate-variants of gated recurrent unit (GRU) neural networks. In Proceedings of the 2017 IEEE 60th international midwest symposium on circuits and systems (MWSCAS). IEEE, 2017, pp. 1597–1600.
. Graves, A.; Graves, A. Long short-term memory. Supervised sequence labelling with recurrent neural networks 2012, pp. 37–45.
- All Figures (1-7, particularly 8-10, but also 11-13) show “Output SIR...” – not “SIR...” (since there is “input SIR”). Please, write in their legends “Output SIR…”.
>>> The legends of the aforementioned figures have been fixed, as well as their captions.
- The paper has one author, while use plural “the authors” at several places in the text.
>>> These sentences refer to the work of the authors of the original papers. This was clarified throughout the manuscript.
- "where..." after equations (1) and (2) should not be indented like regular paragraphs.
>>> Fixed.
- ".)" -> ")."
>>> Fixed
- was calculated was recorded (?)
>>> Modified to "was calculated and recorded".
- "(16 kHz.)" -> "(16 kHz)"
>>> Fixed.
Reviewer 4 Report
The authors have presented three methods to improve speech enhancement. This has a significant contribution to the related research fields. Specifically, speech enhancement using deep learning is a new concept. The paper is well-written with pretty much details. However, the authors must include the following information in the paper to make it more complete.
(a) The major limitation of the paper is that it lacks information regarding the database. The authors only provided the link to the data source. But, they did not offer many details on it. The authors are required to provide more details on the database.
(b) The authors have used three models namely Spectral Feature Mapping with Mimic Loss, MetricGAN+, and Demucs-Denoiser. The authors did not provide enough details on selecting these three methods as there are other methods available in the literature.
(c) The authors have used the SpeechBrain framework. They are requested to provide more details on the SppechBrain as this is the core of their proposed methods. At least they must provide some model for the SpeechBrain.
Author Response
The author greatly appreciates the comments from the reviewer. Here is a point-by-point response to the reviewer's comments:
- The major limitation of the paper is that it lacks information regarding the database. The authors only provided the link to the data source. But, they did not offer many details on it. The authors are required to provide more details on the database.
>>> The following text was added to expand on the Librispeech corpus description: " The LibriSpeech corpus is a set of recordings of users reading novels, as part of the LibriVox project \cite{kearns2014librivox}. It bears several sub-sets of recordings, aimed to be used for training and validating speech-related applications. The `dev-clean' subset is used for validation, and it has little to no noise or perturbations, which makes them an appropriate point of reference from which the corpus can be based on."
- The authors have used three models namely Spectral Feature Mapping with Mimic Loss, MetricGAN+, and Demucs-Denoiser. The authors did not provide enough details on selecting these three methods as there are other methods available in the literature.
>>> The selection criteria of the techniques to evaluate is presented at the start of Section 2.
- The authors have used the SpeechBrain framework. They are requested to provide more details on the SppechBrain as this is the core of their proposed methods. At least they must provide some model for the SpeechBrain.
>>> As described in the manuscript, the only parts of the SpeechBrain framework that were used in this work were the implementations of the perturbations to recordings (evaluation, babble, and noise), since these were employed to create the evaluation corpus. No model was trained using SpeechBrain framework. Thus, this author does not believe it is necessary to detail its explanation (further than the already-presented explanation of the implementation of the aforementioned perturbations) since doing so will lenghten an already long manuscript with little benefit to the reader.
Reviewer 5 Report
The paper is well organized; the structure is clear to the reader. The author should explain why "the author writes" is sometimes used and sometimes "authors" How was it in real?
Sometimes shortcuts should be expanded, for example, ReLu.
I recommend adding supplementary data as an example (wav files) of the system's work.
Author Response
The author greatly appreciates the comments from the reviewer. Here is a point-by-point response to the reviewer's comments:
- The author should explain why "the author writes" is sometimes used and sometimes "authors" How was it in real?
>>> These sentences refer to the work of the authors of the original papers. This was clarified throughout the manuscript.
- Sometimes shortcuts should be expanded, for example, ReLu.
>>> This has been fixed throughout the manuscript.
- I recommend adding supplementary data as an example (wav files) of the system's work.
>>> Although the author agrees with this recommendation, adding even more information to an already-lengthy manuscript, will probably be overwhelming to the reader. However, adding them to the github repository will be considered.